# Impact of Nanotechnology from Nanosilica to Mitigate N and P Deficiencies Favoring the Sustainable Cultivation of Sugar Beet

**DOI:** 10.3390/nano12224038

**Published:** 2022-11-17

**Authors:** Lívia Tálita da Silva Carvalho, Renato de Mello Prado, José Lucas Farias Da Silva, Patrícia Messias Ferreira, Renan Izildo Antonio

**Affiliations:** Department of Agricultural Sciences, Faculty of Agricultural and Veterinary Sciences (FCAV), São Paulo State University (UNESP), Jaboticabal 14884-900, São Paulo, Brazil

**Keywords:** nanoparticles, nutrition, nutritional disorders, sustainable agriculture

## Abstract

This research aimed to study the effects of the nanosilica supply on Si absorption and the physiological and nutritional aspects of beet plants with N and P deficiencies cultivated in a nutrient solution. Two experiments were performed with treatments arranged in a 2 × 2 factorial scheme in randomized blocks with five replications. The first experiment was carried out on plants under a N deficiency and complete (complete solution with all nutrients), combined with the absence of Si (0 mmol L^−1^) and the presence of Si (2.0 mmol L^−1^). In the other experiment, the plants were cultivated in a nutrient solution with a P deficiency and complete, combined with the absence (0 mmol L^−1^) and the presence of Si (2.0 mmol L^−1^). The beet crop was sensitive to the N and P deficiencies because they sustained important physiological damage. However, using nanosilica via fertigation could reverse the damage. Using nanotechnology from nanosilica constituted a sustainable strategy to mitigate the damage due to a deficiency in the beet crop of the two most limiting nutrients by optimizing the physiological processes, nutritional efficiency, and growth of the plants without environmental risks. The future perspective is the feasibility of nanotechnology for food security.

## 1. Introduction

Most cultivated tropical soils may have source material with low nutrient levels associated with a low organic matter content and an acid reaction. The advance of climate change with periods of excess rainfall and others with water deficits can reduce the availability of nutrients in the soil, worsen the deficiency, and impair crop growth and productivity, such as with the beet crop (*Beta vulgaris* L.).

This species has a high economic importance, being used as a sugar or vegetable source [1]. N and P are the most limiting crop nutrients in different regions of the world.

A nitrogen deficiency in beet cultivation causes a loss of nutritional quality of the roots and leaves and a morphological modification associated with a size reduction [2], mainly due to its role in the structure of chloroplasts and photosynthesis as it is a constituent of the chlorophyll molecule as well as proteins, nucleic acids, and coenzymes, among others [3]. At the same time, a P deficiency promotes losses in the energy metabolism of the plant, decreasing the enzymatic activity, protein synthesis, and signaling of carbohydrate metabolism [4], compromising the development of the roots and the aerial part [5]. These nutritional deficiencies cause morphological changes, translating them into characteristic symptoms [5] that progress to severe ones, causing leaf necrosis and plant death. It is necessary to search for sustainable alternatives such as Si that can mitigate the effects of these nutritional disorders.

These nutritional deficiency facts have led to the need to develop ecologically correct economic strategies, mainly to keep up with the demands of the growing population. Therefore, nanoparticles are opening a new chapter in sustainable crop production [6]. These authors added that applying nanoparticles improves growth and stress tolerance in plants. Thus, information has emerged indicating that silicon (Si) is a beneficial element that can be used as a strategy to improve plant development by reducing different abiotic stresses without environmental risks [7] such as nutritional deficiencies [8]. However, in beet, these effects are unknown [9].

There are reports in other species that supplying Si from conventional sources such as sodium silicate alleviated N deficiency effects in quinoa [10] and forage plants [11]. It also alleviated P deficiency effects in sorghum [12]. The beneficial effect of Si is attributed to its action in the activation of antioxidant defense systems by increasing the synthesis of phenols and carotenoids [13] as well as ascorbic acid [14], favoring membrane integrity [15] and the protection of the photosynthetic system of the plant [15] when decreasing the degradation of chlorophylls [16]. It can also increase nutrient absorption [17].

Several studies have demonstrated the beneficial effects of Si, especially using conventional sources such as potassium silicate, sodium silicate, or calcium silicate in various crops [18]. However, in the beet crop, studies are very limited and restricted to only the foliar application of the element with conventional sources in unstressed plants [19,20,21]. Thus, another form of Si application would be via fertigation with a root application, which could increase the absorption of the element as it would be continuously supplied during the crop cycle, unlike the foliar application that occurs at a given time and growing season [18]. A recent report highlighted the benefits of Si when provided via the radicular versus the foliar route, as seen in maize plants [22].

It was evidenced that most of the indicated studies on Si were not carried out using nanotechnology such as nanosilica. Nanosilica consists of silicon dioxide (SiO_2_) nanoparticles in a colloidal dispersion, with physicochemical characteristics different from those of non-nanoparticulate matter due to its small size (usually less than 100 nm), varied shapes, and large surface area [23]. Using nanosilica can modify the biochemistry and physiology of a plant and, consequently, the agronomic benefits in several species such as rice plants [24], sugar cane [25], sorghum [26], cotton [27], and tomato [28]. However, there is a need for further research to prove this in other species, including vegetables.

Thus, if it is considered that the beet crop is sensitive to N and P deficiencies because they result in stress and physiological disorders and that by applying Si in the form of nanosilica via fertigation can promote a significant increase in Si absorption, it can strengthen the following hypothesis. Using Si can mitigate N and P deficiencies in a beet crop by promoting benefits in the antioxidant defense system, possibly with increased antioxidant compounds favoring the physiological and nutritional aspects of the plant.

Therefore, in this research we aimed to study the effects of a nanosilica supply on the Si absorption, antioxidant compound production, extravasation of cellular electrolytes, and physiological and nutritional aspects of beet plants with N and P deficiencies cultivated in a nutrient solution.

Suppose the hypothesis of this research is accepted. In that case, it will be possible to understand the action mechanisms of nanosilica in mitigating the effects of N and P deficiencies. It should strengthen the sustainable cultivation of sugar beet without environmental risks or global implications because there are many regions of vegetable cultivation in areas with a deficiency of one or two of the most limiting macronutrients of this species.

## 2. Material and Methods

### 2.1. Location of Experiments

Two experiments were carried out in a greenhouse at UNESP Campus de Jaboticabal from July to October 2021 using the cultivate ‘Early Wonder’ beet. During the experimental period, the temperature and relative humidity data were recorded inside the greenhouse with a thermo-hygrometer (U23-001, Sigma Sensors, São Paulo/SP, Brazil) (Figure 1).

### 2.2. Treatments and Experimental Design

Two experiments were performed with treatments arranged in a 2 × 2 factorial scheme in randomized blocks with five replications. The first experiment was conducted on plants under a N deficiency and complete (complete solution with all nutrients), combined with the absence of Si (0 mmol L^−1^) and the presence of Si (2.0 mmol L^−1^). In the other experiment, the plants were grown in a nutrient solution with a P deficiency and complete, combined with the absence (0 mmol L^−1^) and presence of Si (2.0 mmol L^−1^). The nanosilica Bindzil (AkzoNobel^®^, Rio de Janeiro/RJ, Brazil) was used for the two experiments (Si: 168.3 g/L; specific superficial area: 300 m^2^/g; pH: 10.5; density: 1.2 g cm^−3^; Na_2_O: 0.5%; viscosity: 7cP).

The nutrient solution used was that of Hoagland and Arnon [29], with a modification of the iron source to Fe-EDDHA.

Sowing was performed directly in a polypropylene pot with a 1.7 dm^3^ capacity filled with 1.5 dm^3^ of medium-texture washed sand; two seeds per pot were used. Five days after an emergence, the plants were fed daily with a supply of the complete nutrient solution, with 100 mL of the solution supplied with an ionic strength of 10% of the concentration of the original solution for ten days. From this period on, the omission of nutrients (−N and −P) began according to each experiment and the concentration of the solution was gradually increased until it reached 75%, maintaining it until the end of the experiment. The pH value of the nutrient solution was adjusted to 5.5 ± 0.2 using a solution of 1.0 mol L^−1^ of hydrochloric acid or sodium hydroxide. The substrate was leached weekly by applying 500 mL of deionized water per vessel to avoid salinization. The nutrient solution was applied to the experiment vessels after 2 h of leaching.

The plants were collected 90 days after emergence and the evaluations described below were performed.

### 2.3. Analysis

#### 2.3.1. Nitrogen and Phosphorus Use: Efficiency and Accumulation

The plants were separated into shoots and roots and washed with a detergent solution (0.1% *v*/*v*; 1 mL/1000 mL), a HCl solution (0.3% *v*/*v*; 3 mL/1000 mL), and deionized water. The plant tissue samples were then dried in an oven with forced air circulation (TE394/3-MP, Tecnal, Piracicaba/SP, Brazil) (65 ± 5 °C) until a constant mass was reached and then ground. After that, a chemical analysis was performed to determine the N content from an extraction with concentrated sulfuric acid, followed by distillation and titration with sulfuric acid [30]. The P content was determined by digesting the samples of the plant material using a digestive mixture of perchloric and nitric acid, according to the methods described by Bataglia [30].

The calculation of N and P accumulations in the shoot was performed from the product of the content (g kg^−1^) x shoot dry mass. With the data of the accumulation of each nutrient in the aerial part of the plant, we calculated the use efficiencies of N and P as: (dry matter of the aerial part)^2^/accumulation of nutrients in the aerial part [31].

#### 2.3.2. Silicon Accumulation

The Si content was determined by digestion according to Kraska [32] and read according to the methodology described by Korndörfer [33] in a spectrophotometer (B442, Micronal, Santo André/SP, Brazil) at 410 nm. The silicon accumulation in the shoot was calculated from the content (g kg^−1^) × dry mass.

#### 2.3.3. Dry Mass of Plants

The dried plant material was weighed using an analytical balance (accuracy of 0.001 g) to obtain the dry mass of the shoots and roots of the plants.

#### 2.3.4. Photosystem II Quantum Efficiency (Fv/Fm)

Photosystem II (PSII) quantum efficiency photosynthetic measurements were taken between 7 am and 8 am on the first fully grown sheet using a portable fluorometer (Os30P+, Opti-Sciences Inc., Hudson, NH, USA) [34].

#### 2.3.5. Electrolyte Extravasation Index

Five disks (26.4 mm^2^ each) were collected from the first fully developed leaf and then immersed in deionized water for 2 h. The initial electrical conductivity (EC_1_) of the solution was read using a conductivity meter (AK51, Akso, São Leopoldo/RS, Brazil). The samples were autoclaved at 121 °C for 20 min and the final electrical conductivity (EC_2_) was determined after cooling. The electrolyte extravasation index was then calculated following the method described by Dionisio-Sese and Tobita [35].

#### 2.3.6. Determination of Total Phenol Contents

The total phenol contents were determined in the leaves according to the method proposed by [36]. Readings were performed in a spectrophotometer (B442, Micronal, Santo André/SP, Brazil) at 765 nm.

#### 2.3.7. Quantification of Chlorophyll and Carotenoid Pigments

The quantification of chlorophyll and carotenoid pigments in the fresh leaf samples was performed according to the methodology proposed by Lichtenthaler [37]. The absorbances were read at 663 nm for chlorophyll a (Chl a), 647 nm for chlorophyll b (Chl b), and 470 nm for carotenoids (Chl x + c) using a spectrophotometer (Beckman DU 640, East Lyme, CT, USA). The absorbance values were introduced into Equation (1) for the determination of chlorophyll a, Equation (2) for chlorophyll b, and Equation (3) for carotenoids:c_a_ = ((12.25·(d·A_663_)) − (2.79·(d·A_647_)))·m(1)
where d is the sample dilution, A_663_ is the reading value at an absorbance of 663 nm, A_647_ is the reading value at an absorbance of 647 nm, and m is the mass of the sample.
c_b_ = ((21.5·(d·A_647_)) − (5.1·(d·A_663_)))·m(2)
where d is the sample dilution, A_647_ is the reading value at an absorbance of 647 nm, A_663_ is the reading value at an absorbance of 663 nm, and m is the mass of the sample.
x + c = (((1000·(d·A_470_)) − (1.82·c_a_) − (85.02·c_b_))/198)·m(3)
where d is the sample dilution, A_470_ is the reading value at an absorbance of 470 nm, c_a_ is the result of Equation (1), c_b_ is the result of Equation (2), and m is the mass of the sample.

### 2.4. Statistical Analysis

The data were checked for normality (the Shapiro–Wilk test) and homogeneity of variances (Levene’s test) and then subjected to an analysis of variance using the F test (*p* ≤ 0.05). The Tukey test (*p* ≤ 0.05) compared the mean values of the treatments. The statistical analyses were performed using SAS^®^ statistical software, single version (Cary, NC, USA).

## 3. Results

The plants from the complete treatment compared with those from the treatments with N and P deficiencies showed a greater accumulation of N and P in the shoot and the root, both in the absence and presence of Si (Figure 2a–d). The Si supply in the plants with complete and N and P deficiencies increased the N and P accumulations in the shoot and root (Figure 2a–d).

There was a greater Si accumulation in the shoots and roots of the beet plants in the complete treatment in the absence or presence of Si compared with the plants grown under N and P deficiencies in the nutrient solution (Figure 2e,f). Regarding its absence, the Si supply in the nutrient solution promoted a greater Si accumulation in all the treatments studied.

It was observed that there was a greater N use efficiency in the control treatment plants than in the N-deficient treatment plants in the absence and presence of Si (Figure 2g). However, the treatment deficiency in P compared with the plants of the complete treatment in the absence or presence of Si provided a greater P use efficiency (Figure 2h). It was also observed that the Si supply in the nutrient solution provided a greater use efficiency in all the treatments studied (Figure 2g,h).

Compared with the other treatments in the absence and presence of Si, the complete treatment plants provided higher levels of total phenols (Figure 3a). However, the supply of Si via the nutrient solution increased the total phenol content in all treatments (Figure 3a).

The carotenoid production in the complete treatment plants was higher than in the other treatments, both in the absence and presence of Si (Figure 3b). However, it was observed that the Si supply in the nutrient solution increased the carotenoid production in the plants with a deficiency of the two nutrients studied (Figure 3b).

There was a lower rate of extravasation of cellular electrolytes in the complete treatment plants compared with the plants under N or P deficiencies, regardless of the Si supply (Figure 3c). There was a decrease in cellular electrolyte extravasation when there was a Si supply via a nutrient solution in all treatments compared with the plants lacking Si (Figure 3c).

Compared with the plants deficient in N or P, the complete treatment plants showed a higher production of total chlorophyll, both in the absence and presence of Si (Figure 3d). Furthermore, it was observed that there was a greater production of total chlorophyll when Si was supplied in the nutrient solution in all the treatments of this study (Figure 3d).

It was also verified that, in the plants of the complete treatment, there was no difference regarding the supply of Si in the PSII quantum efficiency (Figure 3e). However, in the plants with N or P deficiencies, there was an increase in the PSII quantum efficiency with the presence of Si compared with its absence in the nutrient solution (Figure 3e).

Compared with the plants in the treatments with N and P deficiencies, the complete treatment plants showed a higher shoot and root dry mass production, both in the absence and presence of Si (Figure 4a,b). Regarding its absence, the Si supply increased the dry mass production in the shoot and root in the complete treatment plants and the N- and P-deficient plants (Figure 4a,b).

## 4. Discussion

In beet plants, nitrogen (-N) and phosphorus (-P) deficiencies in the nutrient solution led to a decrease in these elements due to low N and P absorption, which caused biological damage to the plants. The N and P deficiencies promoted an imbalance in the nutrient absorption [38], compromising the nutritional roles of these elements in the plants. Nitrogen has a structural function constituting the chlorophyll molecule that plays a fundamental role in transferring excitation energy to photosystems [39]. P is also linked to metabolic activities, from energy transfer as a constituent of NADP (nicotinamide adenine dinucleotide phosphate) and ATP (adenosine triphosphate) [40] to a high activity of the enzyme ribulose 1,5-bisphosphate carboxylase (Rubisco). Its low supply directly affects the gas exchange and photosynthetic rate [41], decreasing new cell formation and cell elongation [42].

Thus, it is common to see disturbances in the physiological mechanisms of plants with N and P deficiencies [43] due to the decrease in the levels of chlorophyll and carotenoid pigments, which leads to a reduction in the PSII quantum efficiency and, consequently, a reduction in the dry mass of the plants, as observed in this research with beet plants.

Silicon is a sustainable strategy to mitigate N and P deficiencies of plants. It is interesting because this beneficial element is already known as a great mitigator of different stresses, whether biotic [9,44] or abiotic [45,46]. However, there is little information on plants grown without stress [47]. In the beet crop, studies are restricted to a foliar application of Si from conventional sources, which has promoted improvements in crop growth and productivity [48]. Studies are incipient with the application of the element in innovative sources from nanosilica nanotechnology, with application via fertigation, especially when evaluating plants deficient in N and P.

The importance of Si in the uptake of N and P in plants deficient in these nutrients was observed when the nutrients were supplied at the beginning of the experiment and especially in the N and P use efficiency. The effect of Si in increasing the use efficiency of these nutrients drew attention, constituting a good indication that this element favors the ability of the plant to use N and P in important nutritional functions involved in the physiological processes responsible for dry mass production.

In addition to the nutritional benefits of Si in plants deficient in N and P, its antioxidant action became clear from the increased levels of phenols and carotenoids. Silicon acts on the metabolism of these phenolic and carotenoid compounds, favoring their synthesis [49], which can be maximized due to the Si enzymatic action increasing the activation of the enzymes involved in the secondary defense metabolism [50]. This increase in the phenol content in the plant, mediated by Si, acts on the balance of the antioxidant system, directly eliminating active molecular oxygen and H_2_O_2_, inhibiting lipid peroxidation [51], and avoiding cellular compound degradation.

This study proved that Si decreased the electrolyte leakage rate in plants with N and P deficiencies, reinforcing its modulation of antioxidant production, thus preventing membrane damage and cytosol efflux into the free space between cells.

Another benefit of Si is that the increase in antioxidant compounds and the decrease in oxidative stress reduces chlorophyll degradation [13]. Therefore, the effect of Si delaying this process benefits plants with N and P deficiencies, as evidenced by the increase in the Fv/Fm values; that is, the quantum efficiency of photosynthesis.

Overall, it became evident that an increase in the Si accumulation by the plant by increasing the photosynthetic pigments and the photosynthetic efficiency favored the ability of the plant to convert dry mass, favoring the dry mass increase and proving that the nanosilica alleviated the N and P deficiencies in beet plants. This effect of Si on the plant physiological aspects has also been recently reported in other species such as quinoa [10] and forager plants [52].

The results of this study allow us to accept the hypothesis that using nanosilica can mitigate N and P deficiencies in beet crops by promoting benefits in the antioxidant defense system, possibly with an increase in the antioxidant compounds favoring the physiological and nutritional aspects of the plant.

It should also be noted that the beneficial effect of nanosilica was evidenced even in plants cultivated without a nutritional deficiency. This effect of Si occurred due to the improvement of the antioxidant capacity, demonstrated by the increase in the production of phenolic compounds, which, in turn, reduced the electrolyte leakage rate. It favored an increase in the total chlorophyll and carotenoid levels, even though it did not affect the PSII quantum efficiency. However, there was an increase in the dry matter production.

The results of this study open the way for nanosilica nanotechnology to enable the sustainable cultivation of beet for marginal regions in low fertility soils or that employ sub-doses of nitrogen and phosphate fertilizers typical of the low-technology agriculture developed in many underdeveloped countries with limited financial resources. The perspective is that further studies will be conducted on this vegetable with nanosilica in field crops, which may include K, Ca, Mg, and S in addition to the nutrients studied, as it may increase benefits in beet production in different regions of the world.

## 5. Conclusions

A beet crop was sensitive to N and P deficiencies because it sustained important physiological damage. However, using nanosilica via fertigation could reverse the damage. Using nanosilica nanotechnology constituted a sustainable strategy for beet cultivation regarding the two most limiting nutrients of the crop by optimizing the physiological processes, nutritional efficiency, and growth of the plant without any environmental risks.

The future perspective is the feasibility of nanotechnology for food security, enabling the production of vegetables in nutrient-poor environments.

## Figures and Tables

**Figure 1 nanomaterials-12-04038-f001:**
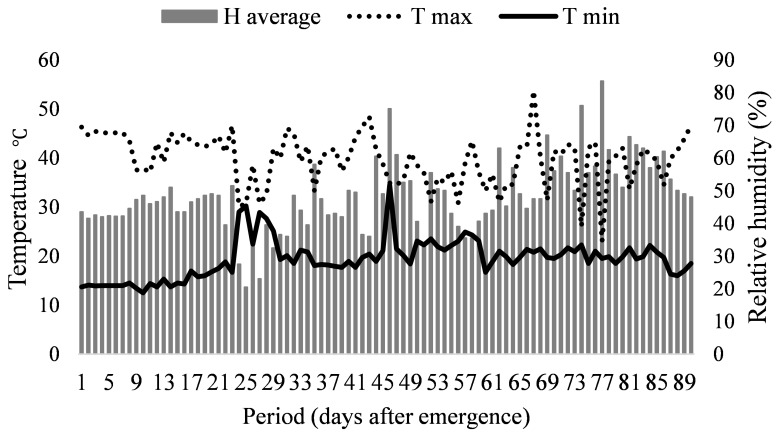
Maximum temperature, minimum temperature, and average relative humidity inside the greenhouse during the experiments.

**Figure 2 nanomaterials-12-04038-f002:**
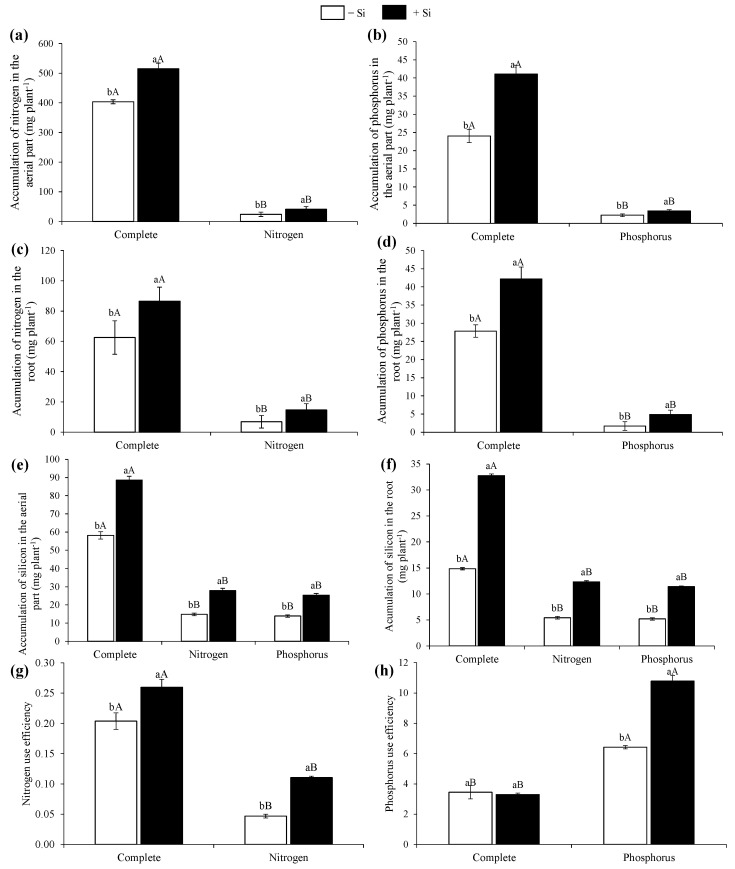
Accumulation of nitrogen (**a**,**c**), phosphorus (**b**,**d**), silicon (**e**,**f**), and nitrogen and phosphorus use efficiency (**g**,**h**) in beet plants with complete nutrition and nitrogen or phosphorus deficiencies in the presence and absence of silicon. Lowercase letters demonstrate differences from Si within the same nutritional status and uppercase letters demonstrate differences from the full treatment according to Tukey’s test at a 5% probability.

**Figure 3 nanomaterials-12-04038-f003:**
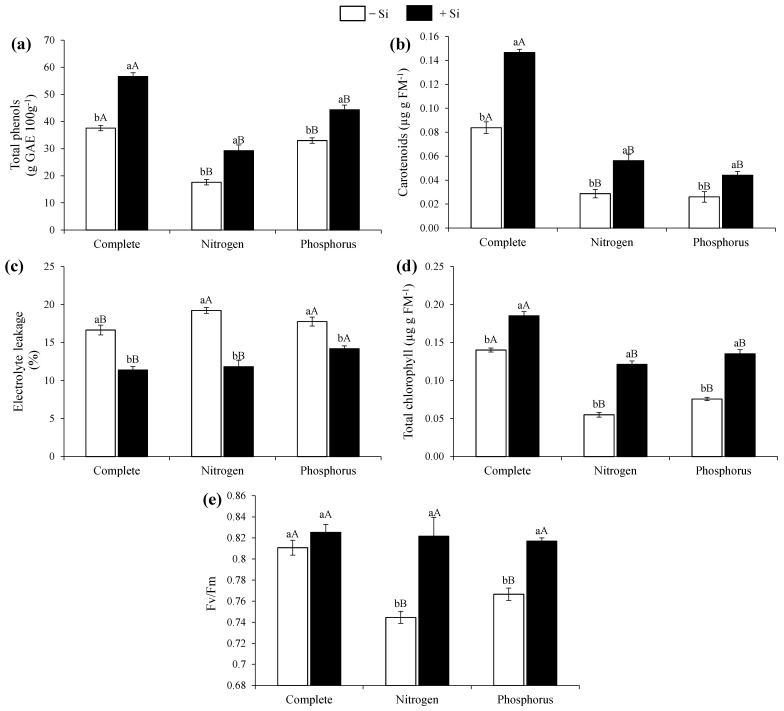
Total phenols (**a**), carotenoids (**b**), electrolyte leakage (**c**), total chlorophyll (**d**), and Fv/Fm (**e**) in beet plants with complete nutrition and nitrogen or phosphorus deficiency in the presence and absence of Si. Lowercase letters demonstrate differences from Si within the same nutritional status and uppercase letters demonstrate differences from the full treatment according to Tukey’s test at a 5% probability.

**Figure 4 nanomaterials-12-04038-f004:**
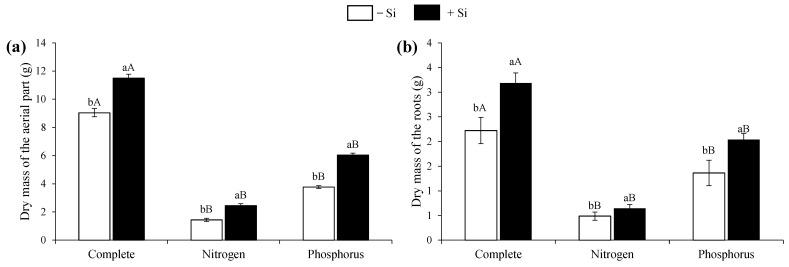
Dry mass of the aerial part (**a**) and roots (**b**) of beet plants with complete nutrition and nitrogen or phosphorus deficiencies in the presence and absence of Si. Lowercase letters demonstrate differences from Si within the same nutritional status and uppercase letters demonstrate differences from the full treatment according to Tukey’s test at a 5% probability.

## Data Availability

The data presented in this study are available on request from the corresponding author.

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
