# Peer review of "Impact of Nanotechnology from Nanosilica to Mitigate N and P Deficiencies Favoring the Sustainable Cultivation of Sugar Beet"

_nanomaterials, 2022, doi:10.3390/nano12224038_

Round 1
Reviewer 1 Report
This could be an interesting paper with possible applications, but more details regarding methodology are necessary.
The phrases from lines 61-65 are not clear. Please revise these.
Please mention if nanosilica it was prepared by authors or it was commercial? Give more details about this.
Please give more details in Section 2.2 regarding treatments and experimental design. For example: the number of seeds / pot, the volume (quantity) of nutrient used, the periodicity of using the nutrient or if it was applied only once.
In the section 2.2.1, line 127, please mention the plant drying temperature.
Line 137: it is not clear about what equation is talking. "(dry matter of the aerial part)2/ accumulation of nutrients in the aerial part" is not an equation. Please re-write this this part.
Line 151: "mm2" must be with superscript "mm2".
Lines 162 - 164: please introduce the equation used for chlorophylls. The phrase "Extract readings were obtained ... DU 640 spectrophotometer" must be changed because it is unclear and incorrect. The absorbances were read at 663nm and 647nm. The values of absorbances were introduced in some equations and the quantities of chlorophylls were determined in this manner.
Lines 209 - 212: please revise these phrases because you use FSII and PSII, but I suppose that in both cases is PSII. Also, I see that the FSII is higher in all cases with Si than without Si.
Line 237: please explain the acronym: NADP and ATP.
All references must be cited according to instructions for authors.
Reviewer 2 Report
The manuscript“Impact of Nanotechnology from Nanosilica to Mitigate N and P Deficiency Favoring Sustainable Cultivation of Sugar Beet”, by LíviaTálita da Silva Carvalho, Renato de Mello Prado, José Lucas Farias Da Silva, Patrícia Messias Ferreira and Renan Izildo Antonio, submitted for publication in the Journal Nanomaterials, section Environmental Nanoscience and Nanotechnology, Special Issue State-of-the-Art Environmental Nanoscience and Nanotechnology represents an original study of the effects of the nanosilica supply on Si absorption, antioxidant compound production, extravasation of cellular electrolytes, and the physiological and nutritional aspects of beet plants with N and P deficiency cultivated in a nutrient solution, declared by the authors.
The results reported are of substantial fundamental and applied interest as they are closely related to the ability of beet crop production in specific conditions and especially in the presence of nutrient deficit, which is the case of tropical type of soil.
However, I have the following remarks and comments:
1. Some abbreviations could be presented with the full names when met for the first time in the text: ex. UNESP, NADP, ADP etc.
2. Тhe letters in the labels of all the figures are blurred and could be replaces with better resolution of the figures.
3. The introduction of the labels with lowercase and uppercase letters seems meaningless (if I have understand correctly the scheme), as the presence or absence of Si is expressed by the color of bars (white or black) and the nutrition regime is noted below each bar.
4. The experimental errors should be estimated and noted on the bars in the figures. Some conclusions are unconvincing because of its lack.
Some typos and small inexactitudes are also detected:
Line 95: “cultivar”
Reviewer 3 Report
The article describes the use of nanosilica to mitigate N and P deficiency favouring sustainable cultivation of sugar beet.
The theme is interesting, although some concerns must be addressed:
- The abstract must re-written. For the example, the first phrase is too long, its sense being lost. Also, “-1” must be superscript.
- Please use either Present Tense or Past Tense, not a combination. Please verify this throughout the entire manuscript.
- The state-of-the-art must be updated and more references must be introduced.
- Please do not use the personal language in the manuscript (“we”….), but the impersonal diathesis.
- Please provide full information about used reagents and instruments (company, city, country).
- All figures must be included after their first mention in the text.
- Figure 1 must be re-done in order to increase its resolution.
- Please mention in the manuscript the source of nanosilica, not only its properties.
- Lines 126-127: Please mention the exact quantities and volumes for shoots, roots and detergent solution.
- Lines 135-138: please rephrase: “And with the data of the accumulation of each nutrient in the aerial part of the plant, the calculation of the use efficiencies of N and P was conducted according to the equation: (dry matter of the aerial part)2/accumulation of nutrients in the aerial part [31]”.
- Please revise figure 2 in order to increase the fonts size.
- Same comment for figures 3 and 4.
- Please try to verify the entire manuscript so that it does not appear as a technical report, but a scientific material. At this moment, the manuscript seems telegraphic.
- It is mandatory that the English language is revised by a native English speaker.
Round 2
Reviewer 1 Report
Lines 168, 171, 174: please re-write the equations, introducing ca instead of "Equation 1". The equations must be written as follows:
ca = ((12.25·(d·A663))-(2.79·(d·A647)))·m (equation 1)
In the same manner please modify the equation 2 and 3. In equation 3 must appear ca instead of "1" and cb instead of "2".
The figure 2 is cut.
Author Response
1 The equations have been rewritten
2 The figure has been revised
Reviewer 3 Report
Please revise figures 2 and 3 - they are out of the manuscript content!
Author Response
The figures have been revised.